# Are clinically unimportant findings qualified as benign in lumbar spine imaging reports? A content analysis of plain X-ray, CT and MRI reports

Caitlin Farmer *, Romi Haas, Jason Wallis, Denise O'Connor , Rachelle Buchbinder

Musculoskeletal Health and Sustainable Healthcare Unit, School of Public Health and Preventive Medicine, Monash University, Melbourne, Australia

* caitlin.farmer1@monash.edu

## Abstract

### Background

Lumbar spine diagnostic imaging reports may cause patient and clinician concern when clinically unimportant findings are not explicitly described as benign. Our primary aim was to determine the frequency that common, benign findings are reported in lumbar spine plain X-ray, computed tomography (CT) and magnetic resonance imaging (MRI) reports as either normal for age or likely clinically unimportant.

### Methods

We obtained 600 random de-identified adult lumbar spine imaging reports (200 X-ray, 200 CT and 200 MRI) from a large radiology provider. Only reports requested for low back pain were included. From the report text, one author extracted each finding (e.g., 'broad-based posterior disc bulge') and whether it was present or absent (e.g., no disc bulge) until data saturation was reached, pre-defined as a minimum of 50 reports and no new/similar findings in the last ten reports within each imaging modality. Two authors independently judged whether each finding was likely clinically unimportant or important. For each likely clinically unimportant finding they also determined if it had been explicitly reported to be benign (expressed as normal, normal for age, benign, clinically unimportant or non-significant).

### Results

Data saturation was reached after coding 262 reports (80 X-ray, 82 CT, 100 MRI). Across all reports we extracted 3,598 findings. Nearly all reports included at least one clinically unimportant finding (76/80 (95%) X-ray, 80/82 (98%) CT, 99/100 (99%) MRI). Over half of the findings (n = 2,062, 57%; 272 X-Ray, 667 CT, 1123 MRI) were judged likely clinically unimportant. Most likely clinically unimportant findings (90%, n = 1,854) were reported to be present on imaging (rather than absent) and of those only 18% (n = 331) (89 (35%) X-ray, 93 (16%) CT and 149 (15%) MRI) were explicitly reported as benign.

**Data Availability Statement:** "We have uploaded our extracted data to Open Science Framework (available at https://osf.io/ut9am/). Due to our legal agreement with i-Med, this does not include the

original radiology reports. However, requests to access these can be obtained by emailing research@i-med.com.au and then forwarded to the Monash University Human Research Ethics Committee (MUHREC) (muhrec@monash.edu)."

**Funding:** This work was supported by a National Health and Medical Research Council (NHMRC) Program Grant: Using healthcare wisely: reducing inappropriate use of tests and treatments (APP1113532) 2017-21. RB is funded by an Australian National Health and Medical Research Council (NHMRC) L3 Investigator Fellowship (APP1194483) www.nhmrc.org.au The funders had no role in study design, data collection and analysis, decision to publish, or preparation of the manuscript.

**Competing interests:** The authors have declared that no competing interests exist.

## Conclusion

Lumbar spine imaging reports frequently include findings unlikely to be clinically important without explicitly qualifying that they are benign.

## Introduction

Degenerative changes, including disc bulges and facet joint degeneration, are common findings described in lumbar spine imaging reports [1–3]. These changes are increasingly prevalent with age and are equally as common in people with and without low back pain [1]. Longitudinal population-based studies have confirmed that such degenerative changes do not have clinically important associations with either current or future back pain, even when multiple changes are present [4, 5]. Reporting on the presence of degenerative changes in imaging reports without clarifying that they are likely clinically unimportant has the potential to lead to overdiagnosis and overtreatment, and cause patient anxiety about the seriousness and persistence of symptoms [6–9].

There is a paucity of guidance for radiologists on how to communicate findings of limited clinical relevance in a manner that does not alarm the reader. A scoping review that included six radiology reporting guidelines found that three of the guidelines recommended reporting the presence of normal findings, although there was limited advice about how this should be actioned [10]. For example, the Royal Australian and New Zealand College of Radiologists (RANZCR) guideline suggests that normal findings be noted when it would make a difference to the referrer or when absence of such a statement would create ambiguity [11].

In addition, only three guidelines provide guidance on communicating confidence or certainty in reports. The UK College of Radiologists guideline recommends that the level of certainty or doubt surrounding an imaging diagnosis be clearly documented in the report [12], the Canadian College of Radiologists guideline suggests the focus should be on findings that offer potential for resolution of the clinical question [13], while the RANZCR guideline recommends avoiding use of vague modifiers, such as 'possibly represents' [11]. These recommendations are in keeping with other literature that recommends avoiding ambigious statements or hedging vocabulary, such as 'there appears to be...', to minimise confusion for the reader. Using hedging vocabulary such as 'seen' or 'identified' is discouraged as it suggests that something may have been missed; for example 'no fracture seen' is a less certain statement than 'no fracture' [14, 15].

The choice of phrasing in imaging reports can influence management decisions. For example a US study providing hypothetical scenarios to clinicians found they were more likely to request further imaging if an incident 5mm liver lesion was described as 'most likely a cyst' compared to being a 'benign cyst (46% and 2% respectively) [16]. Providing reassuring statements, such as 'findings are normal for age', or avoiding alarming descriptors, such as 'degeneration', 'tear' or 'rupture', have been suggested as possible ways to reduce misinterpretation of clinically unimportant findings in lumbar spine imaging reports [6, 17].

One previous study has investigated the extent to which degenerative changes are reported in lumbar spine imaging reports and how they are described [2]. Based upon examination of 120 consecutive plain X-ray reports requested in primary care, they found that almost three quarters noted the presence of degenerative changes. Only 2% of reports explicitly stated these were normal for age, while 14% indicated the changes were either 'mild' or 'slight', which may be a less explicit way of indicating to the reader that a particular finding is of limited clinical

relevance. Another study performed a content analysis of plain X-Ray and MRI imaging reports of patients with persistent low back pain and explored, through interviews, which terms negatively impacted the patients' perceived prognosis [18]. The terms 'wear and tear' and 'disc space loss' were associated with a significantly worse perceived outcome based upon patients' interpretation that these terms signified the spine was 'deteriorating', 'crumbling', 'collapsing' and/or the discs were 'wearing out'.

The primary aims of this study were to determine (a) the frequency that likely clinically unimportant findings are reported in lumbar spine plain X-ray, CT and MRI reports, and (b) the frequency that they are explicitly reported to be benign (i.e., normal, normal for age, benign, clinically unimportant or non-significant). Second, we investigated the frequency of adjectives (e.g., mild, severe) used to describe these findings and how frequently terms of uncertainty (i.e., vague modifiers, hedging vocabulary) were used.

## Methods

### Study design

We performed a content analysis of a random sample of fully de-identified lumbar spine plain X-ray, CT and MRI imaging reports from iMed, a large radiology service provider in Victoria, Australia.

A random sample of 600 (200 X-ray, 200 CT and 200 MRI) reports written between 1 January 2019 and 30 June 2021 were collected in July 2021 and this study was conducted over the following year. To obtain the random sample for each modality, we used the 'Rand' function in Excel to identify random dates within this time period for each imaging modality. If more than one report was identified for a selected day, we again used the Rand' function in Excel to select another report at random. For days without a report, we used the next randomised date.

A research assistant, not otherwise involved in the study, extracted the complete text of the identified X-ray, CT and MRI reports, including the patient sex, date of birth, date of imaging examination, requesting clinician specialty (e.g., GP, orthopaedic surgeon, rheumatologist), where available, and reporting radiologist using a standardised MS Excel data collection form. To ensure anonymity of the reporting radiologist a unique numerical code was assigned for each radiologist. The de-identified extracted reports were then provided to the research team. No data that could identify patients, referrers or radiologists were provided.

### Eligibility criteria

We included lumbar spine imaging reports for people of any age that indicated that the imaging had been requested for low back or radicular lower limb pain. Reports that covered multiple body regions (e.g., thoracolumbar spine) were included only if the report of the lumbar spine could be clearly separated from reporting on other body regions. We excluded reports of imaging performed following major trauma and those requested to explicitly rule in/out serious causes (i.e. infection, malignancy, fracture), imaging post surgery or imaging performed for monitoring purposes. We also excluded any report that included a serious finding (e.g., vertebral fracture or metastatic disease) regardless of whether the clinical notes queried the presence of such a finding.

### Data extraction

From the text of the reports relevant to the lumbar spine, one author (CF), a physiotherapist with expertise in low back pain, extracted each individual finding including any adjectives describing the finding (e.g., 'mild posterior disc bulge'). This was continued until data

saturation was reached, pre-defined as no new/similar findings in the last ten reports within each imaging modality.

The same author also extracted each term of uncertainty, including vague modifiers and hedging vocabulary based, *a priori*, on published lists of these terms [14–16, 19], and consensus among the authors for additional terms. We grouped terms of uncertainty that had similar meaning together (e.g., 'not shown', 'identified' and 'seen'). A second author (JW or RH, both also physiotherapists with expertise in low back pain) checked all extracted data and differences were resolved by consensus.

For each individual finding, two authors (CF, JW or RH) independently determined whether the finding was likely clinically unimportant or important, based, *a priori*, on published evidence about the relevance of imaging findings, the report context, and/or author team (also included rheumatologist with low back pain expertise (RB) and occupational therapist (DOC)) consensus for equivocal findings. S1 Table provides the list of likely clinically unimportant findings based upon the published evidence. We grouped findings based on anatomical structure (i.e., disc, facet joint, etc.,) and pathology present (e.g., bulge, arthropathy). Findings that described the same or similar abnormality (e.g., 'disc height loss' and 'disc space narrowing') were grouped together.

For findings that are usually considered clinically unimportant (e.g., 'disc protrusion'), if there was evidence within the context of the report of its potential importance (e.g., 'compressing a nerve root'), or if the clinical importance of a finding was ambiguous (e.g., 'mild to moderate canal stenosis'), we erred on being conservative and categorised the finding as likely clinically important. When a clinically unimportant finding was reported to be absent (e.g., 'no disc bulge') we recorded that separately.

For each likely clinically unimportant finding the same two independent authors (CF and RH or JW) recorded whether there was an explicit qualification that the finding was benign. This could have been stated as 'normal', 'normal for age', 'benign', 'clinically unimportant', 'non-significant' and/or other related synonyms (e.g., 'normal alignment' or 'alignment satisfactory'). Any disagreements were resolved by discussion with all authors.

## Sample size and data saturation

The sample size was informed by previous content analyses of lumbar spine imaging reports [2, 18, 20], and the data saturation stopping rule described by Francis et al. [21]. A pilot study investigating the content of lumbar spine imaging performed in patients presenting with back pain to an emergency department of one metropolitan hospital in Victoria, Australia, indicated that a minimum of 50 reports, and likely less than 100 reports, of each modality would be needed [20].

We identified 200 reports for each modality to allow for the various terms known to describe similar radiological findings [22], as well as potential exclusions. We extracted individual findings until data saturation had been reached, defined as a minimum of 50 reports and the point at which no new findings were identified among the last ten reports, coded separately for each imaging modality. Cumulative frequency tables listed each new term until this stopping rule was met.

## Data analysis

We used descriptive statistics to summarise the demographic characteristics of patients that were imaged, reporting radiologists and imaging referrers. We also measured and reported the report word count, median number of findings per report and median number categorised as likely clinically unimportant or important, the proportion of imaging reports with at least one

likely clinically unimportant finding, the proportion that qualified clinically unimportant findings as benign, adjectives used to describe the findings, and terms of uncertainty. The most common likely clinically unimportant findings and most common adjectives and terms of uncertainty were determined for each imaging modality.

### Ethics

This study was approved by the Monash University Human Research Ethics Committee (Approval ID 27959). Individual participant consent was not required due to the de-identified nature of the data.

## Results

Fig 1 presents a flow chart of the report coding process and a summary of the main findings. Data saturation was reached after coding 262 reports (80 X-ray, 82 CT and 100 MRI). Ninety-five reports (17 X-ray, 66 CT and 12 MRI) were excluded for reasons listed in Fig 1, most commonly because the imaging was performed due to trauma. 3598 separate findings were extracted (454, 1139 and 2005 in the X-ray, CT and MRI reports respectively). Of these, 2062 (57%) (272 in X-ray (60%), 667 in CT (59%) and 1123 in MRI (56%)) were judged to be likely clinically unimportant. Most (n = 1854, 90%) were reported to be present rather than reported to be absent, and only a minority (n = 331, 18%) were explicitly reported to be benign.

Patient demographics, requesting clinician and radiologist details and imaging report characteristics by imaging modality are shown in Table 1. There were more women across all three imaging modalities and the median (range) age varied from 50 (15 to 91) years for MRI to 65 (17 to 98) years for CT reports. Most imaging was requested by GPs (50/80 (63%) X-rays, 49/82 (60%) CTs and 57/100 (57%) MRIs). One hundred and five different radiologists reported on the imaging with over 50 different radiologists for each imaging type (52 X-rays, 56 CT scans and 53 MRI scans). Only 10 radiologists reported imaging for all three imaging modalities and 39 contributed only a single report across all modalities. The median number of reports written by each radiologist was one (range for X-ray: 1 to 5, CT: 1 to 3 and MRI 1 to 6)).

X-ray reports had the fewest number of words and findings (median (range): 48 (8 to 167) and 5 (2 to 14), respectively) and MRI scans had the most (median (range): 162 (56 to 355) and 16 (7 to 47), respectively). Nearly all reports included at least one clinically unimportant finding (76/80 (95%) X-ray, 80/82 (98%) CT, 99/100 (99%) MRI), with a median (range) of 3 (1 to 8), 5 (1 to 32) and 9 (1 to 35) per report for X-ray, CT and MRI respectively. Among reports where clinically unimportant findings were present, few reports qualified all of them as benign (13 (16%) X-ray, 8 (10%) CT, 3 (3%) MRI). The majority of reports included at least one term of uncertainty (53/80 (66%) X-Ray, 75/82 (92%) CT and 82/100 (82%) MRI reports).

### Likely clinically unimportant findings, their frequency, and proportion reported to be benign

S1 Table indicates which clinically unimportant findings appeared in at least one report by imaging modality. The most common likely clinically unimportant findings that were reported to be present are shown by modality and in order of frequency in Table 2, together with the proportion that were explicitly reported as benign. Allowing for different sensitivities between imaging modalities, changes to the discs (e.g., disc height loss, disc bulge), facet joint arthropathy and degenerative changes were reported most commonly. Across all modalities these findings were reported to be benign in less than half of the reports that noted their presence. For example, disc height loss was reported to be present in 54%, 54% and 31% of X-Ray, CT and

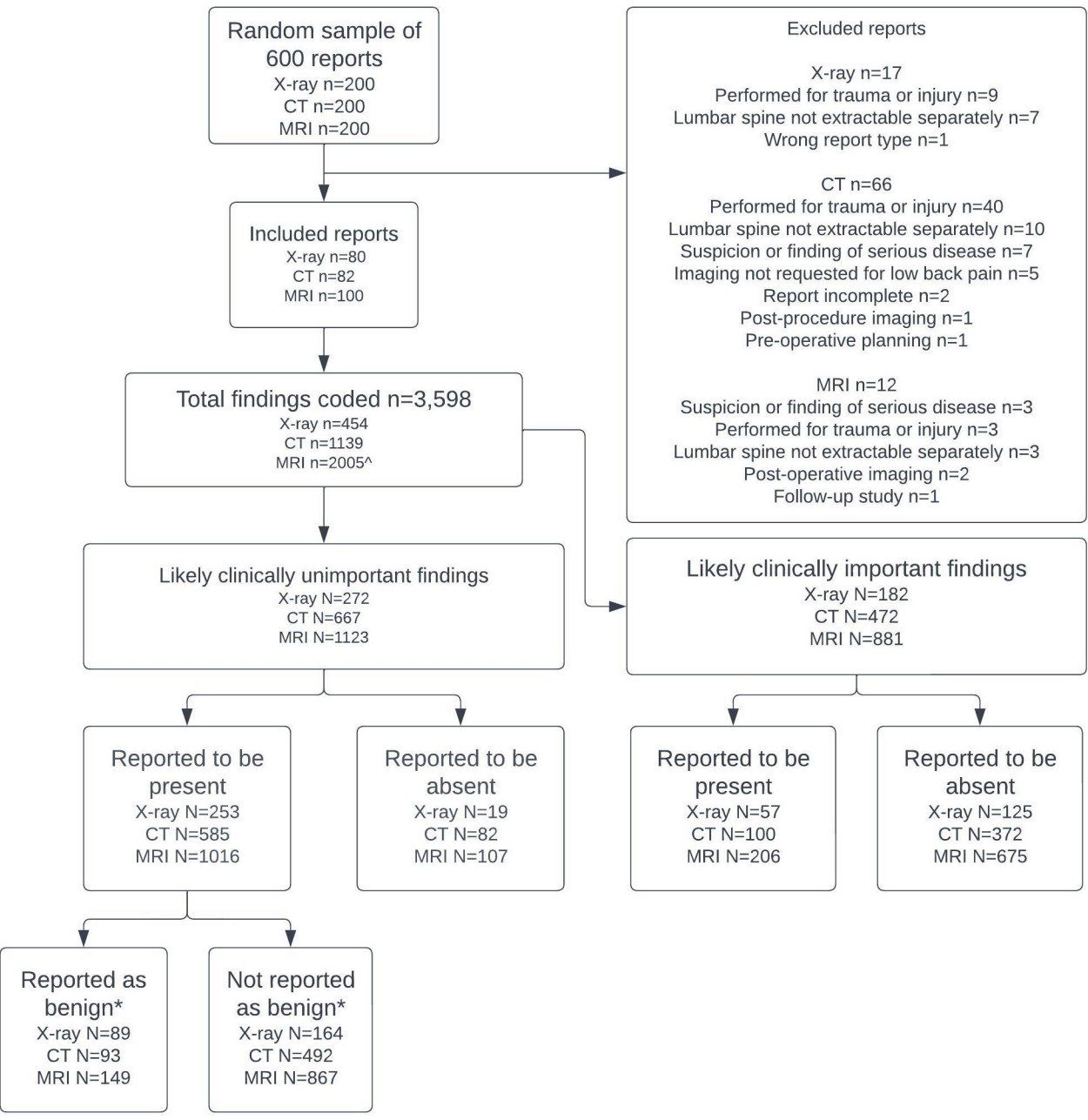

**Fig 1. Flow chart of report coding process.** Notes: *'Benign' findings were explicitly qualified as 'normal', 'normal for age', 'benign', 'clinically unimportant', 'non-significant' and/or other related synonyms. ^One MRI finding ('CBD') was excluded as its meaning was unclear.

MRI reports respectively, and this change was reported to be benign in only 42%, 26% and 33% of those reports respectively. Alignment of segments of the spine, when described (39% X-ray, 40% CT and 46% MRI reports), was most frequently reported to be benign (100% X-Ray, 82% CT and 87% MRI reports that reported the finding).

Likely clinically unimportant findings that were reported to be absent (e.g., 'no disc bulge') were in keeping with those that were reported to be present (S2 Table). S3 Table shows the findings that were considered likely clinically important by modality and in order of

**Table 1. Demographics (sex and age) of patients that had imaging requested, number of reporting radiologists and imaging requestors, and word count and number of imaging findings reported, by imaging modality.**

| | X-ray (N = 80) | CT (N = 82) | MRI (N = 100) |
|---|---|---|---|
| **Patient demographics** | | | |
| Female, N (%) | 49 (61) | 43 (52) | 53 (53) |
| Age in years, median (range) | 56 (6 to 90) | 65 (17 to 98) | 50 (15 to 91) |
| Number of reporting radiologists | 52 | 56 | 53 |
| **Imaging requestor, N (%)\*** | | | |
| General practitioner | 50 (63) | 49 (60) | 57 (57) |
| Emergency physician | 7 (9) | 16 (20) | 14 (14) |
| Chiropractor | 1 (1) | 5 (6) | 4 (4) |
| Physiotherapist | 1 (1) | 1 (1) | 5 (5) |
| Rheumatologist | 2 (3) | 1 (1) | 0 (0) |
| Orthopaedic surgeon | 2 (3) | 0 (0) | 0 (0) |
| Endocrinologist | 2 (3) | 1 (1) | 0 (0) |
| Neurologist | 0 (0) | 0 (0) | 3 (3) |
| Pain physician | 0 (0) | 1 (1) | 4 (4) |
| Unclear | 12 (15) | 6 (7) | 13 (13) |
| **Imaging report characteristics** | | | |
| Word count, median (range) | 48 (8 to 167) | 137 (4 to 389) | 162 (56 to 355) |
| Number of reports that included at least one likely clinically unimportant finding, N (%) | 76 (95) | 80 (98) | 99 (99) |
| Median (range) number of findings per report | 5 (2 to 14) | 11 (2 to 22) | 16 (7 to 47) |
| Median (range) of clinically unimportant findings reported per report | 3 (1 to 8) | 5 (1 to 32) | 9 (1 to 35) |
| Number of reports that qualified all reported clinically unimportant findings as benign, N (%) | 13 (16) | 8 (10) | 3 (3) |
| Mean percent of clinically unimportant findings described as benign per report | 27 | 15 | 14 |
| Number of reports that included at least one term of uncertainty, N (%) | 53 (66) | 75 (92) | 82 (82) |
| Median (range) of terms of uncertainty per report | 1 (1–4) | 3 (1 to 12) | 3 (1 to 14) |

\*One osteopath, one neurosurgeon and one oncologist requested one X-ray, and one urologist and one rehabilitation physician requested a single CT scan.

^ Terms of uncertainty could include vague modifiers and hedging vocabulary

frequency. The most common finding for x-ray was fracture, for CT it was canal stenosis and for MRI it was foraminal stenosis.

## Descriptors of likely clinically unimportant findings and their frequency

Overall, there were 50 different descriptors or groupings of descriptors used to describe likely clinically unimportant findings. The most frequent descriptors, used at least 10 times, overall and by imaging modality are shown in Fig 2. The most common descriptors were used to indicate the severity of the findings (e.g., mild, moderate, severe), and many, shown on the left of the vertical line, indicated a finding was minimal, minor, small or mild. Other commonly used descriptors were 'degenerative' and 'broadbased'.

## Terms of uncertainty and their frequency

Table 3 shows the 22 most frequent groupings of terms of uncertainty. Despite the inclusion of more MRI reports, five groups of uncertain terms contained more terms in the CT modality than either MRI or X-ray, including the most common, 'not shown' (221 occurrences overall; X-ray n = 35, CT n = 104, MRI n = 82). Other groups with more occurrences in CT than X-ray or MRI were 'appear' (53 occurrences overall; X-ray n = 12, CT n = 24 and MRI n = 17), 'may

**Table 2. The likely clinically unimportant findings that were reported to be present by imaging modality: The number (%) of reports that indicated the specific finding was present and frequency (%) of the finding across reports in order of frequency, and the number (%) that were reported to be benign.**

| | Number of reports where finding is present | Frequency of finding across reports | Number of findings reported as benign |
|---|---|---|---|
| **X-ray** | N = 80 reports | N = 253 findings | N = 89 findings |
| | N (%)^ | N (%)# | N (%)α |
| Disc height loss | 43 (54) | 48 (19) | 20 (42) |
| Facet joint arthropathy | 32 (40) | 33 (13) | 4 (12) |
| Lumbar spine alignment | 31 (39) | 31 (12) | 31 (100) |
| SIJ degeneration | 24 (30) | 26 (10) | 17 (65) |
| Scoliosis (mild) | 23 (29) | 24 (10) | 1 (4) |
| General degeneration | 20 (25) | 23 (9) | 3 (13) |
| Osteophytes | 16 (20) | 16 (6) | 1 (6) |
| Lordosis | 11 (14) | 11 (4) | 6 (55) |
| Degenerative disc disease | 10 (13) | 11 (4) | 0 (0) |
| Spondylolisthesis (grade 1) | 9 (11) | 9 (4) | 0 (0) |
| Congenital deformities | 6 (8) | 8 (3) | 0 (0) |
| Pedicles/pars defect | 6 (8) | 6 (2) | 6 (100) |
| Kyphosis (mild) | 2 (3) | 2 (1) | 0 (0) |
| Bone island | 1 (1) | 1 (<1) | 0 (0) |
| Pelvic tilt | 1 (1) | 1 (<1) | 0 (0) |
| Disc protrusion | 1 (1) | 1 (<1) | 0 (0) |
| Bony wedging | 1 (1) | 1 (<1) | 0 (0) |
| Granuloma | 1 (1) | 1 (<1) | 0 (0) |
| **CT (n = 82 reports)** | N = 82 reports | N = 585 findings | N = 93 findings |
| | N (%)^ | N (%)# | N (%)α |
| Facet joint arthropathy | 45 (55) | 88 (15) | 14 (16) |
| Disc bulge | 40 (49) | 73 (13) | 3 (4) |
| Disc height loss | 44 (54) | 59 (10) | 15 (25) |
| Lumbar spine alignment | 40 (40) | 44 (8) | 36 (82) |
| Osteophytes | 23 (28) | 40 (7) | 0 (0) |
| General degeneration | 28 (34) | 36 (6) | 1 (3) |
| Degenerative disc disease | 24 (29) | 32 (6) | 3 (9) |
| Disc protrusion | 17 (21) | 30 (5) | 0 (0) |
| Canal stenosis (mild) | 16 (20) | 24 (4) | 2 (8) |
| Foraminal stenosis (mild) | 12 (15) | 23 (4) | 0 (0) |
| SIJ degeneration | 17 (21) | 17 (3) | 10 (59) |
| Nerve root irritation | 9 (11) | 15 (3) | 0 (0) |
| Ligamentum flavum hypertrophy | 8 (10) | 15 (3) | 0 (0) |
| Vacuum phenomenon | 12 (15) | 13 (2) | 0 (0) |
| Spondylolisthesis (grade 1) | 8 (10) | 13 (2) | 0 (0) |
| Scoliosis (mild) | 10 (12) | 12 (2) | 0 (0) |
| Congenital deformities | 9 (11) | 11 (2) | 2 (18) |
| Lordosis | 9 (11) | 9 (2) | 4 (44) |
| Pars defect | 6 (7) | 8 (1) | 1 (13) |
| Spondylosis | 3 (4) | 6 (1) | 0 (0) |
| Thecal indentation | 5 (6) | 5 (1) | 0 (0) |
| Schmorl's node | 3 (2) | 4 (1) | 2 (50) |
| Disc herniation | 2 (2) | 2 (<1) | 0 (0) |
| Lateral recess stenosis (mild) | 2 (2) | 2 (<1) | 0 (0) |

*(Continued)*

**Table 2.** (Continued)

| | Number of reports where finding is present | Frequency of finding across reports | Number of findings reported as benign |
|---|---|---|---|
| Haemangioma | 1 (1) | 1 (<1) | 0 (0) |
| Kyphosis (mild) | 1 (1) | 1 (<1) | 0 (0) |
| Bone island | 1 (1) | 1 (<1) | 0 (0) |
| Bony wedging | 1 (1) | 1 (<1) | 0 (0) |
| **MRI (N = 100 reports)** | N = 100 reports | N = 1016 findings | N = 149 findings |
| | N (%)^ | N (%)# | N (%)α |
| Disc bulge | 70 (70) | 174 (17) | 5 (3) |
| Facet joint arthropathy | 70 (70) | 152 (15) | 17 (11) |
| Disc desiccation | 44 (44) | 69 (7) | 9 (13) |
| Disc protrusion | 46 (46) | 62 (6) | 1 (2) |
| Foraminal stenosis (mild) | 36 (36) | 61 (6) | 0 (0) |
| Disc height loss | 31 (31) | 58 (6) | 19 (33) |
| Canal stenosis (mild) | 33 (33) | 54 (5) | 9 (17) |
| Nerve root irritation | 33 (33) | 52 (5) | 5 (10) |
| General degeneration | 34 (35) | 50 (5) | 23 (46) |
| Lumbar spine alignment | 46 (46) | 46 (5) | 40 (87) |
| Thecal indentation | 24 (24) | 39 (4) | 0 (0) |
| Annular fissure | 27 (27) | 30 (3) | 0 (0) |
| Disc degenerative disease | 20 (20) | 23 (2) | 1 (4) |
| Ligamentum flavum hypertrophy | 8 (8) | 16 (2) | 0 (0) |
| Scoliosis (mild) | 15 (15) | 15 (2) | 1 (7) |
| Spondylolisthesis (grade 1) | 12 (12) | 14 (1) | 0 (0) |
| Congenital deformities | 11 (11) | 13 (1) | 2 (15) |
| Sacroiliac joint degeneration | 11 (11) | 13 (1) | 8 (62) |
| Lateral recess stenosis (mild) | 11 (11) | 11 (1) | 1 (9) |
| Haemangioma | 10 (10) | 10 (1) | 1 (10) |
| Lordosis | 10 (10) | 10 (1) | 5 (50) |
| Modic changes (Type 2 or 3) | 10 (10) | 10 (1) | 0 (0) |
| Signal change | 7 (7) | 10 (1) | 1 (10) |
| Pars defect | 6 (6) | 6 (1) | 0 (0) |
| Osteophytes | 5 (5) | 6 (1) | 0 (0) |
| Schmorl's node | 5 (5) | 6 (1) | 1 (17) |
| Disc herniation | 4 (4) | 4 (<1) | 0 (0) |
| Tarlov cyst | 1 (1) | 1 (<1) | 0 (0) |
| Bony wedging | 1 (1) | 1 (<1) | 0 (0) |

Note all 'alignment' terms reproduced as written in the report, e.g. alignment, scoliosis, lordosis

^N = number of reports (% of all reports for same modality)

#N = number of times finding present (% of all findings for same modality)

αN = number of times finding reported as benign (% of same finding)

be' (15 occurrences; X-ray n = 2, CT n = 7, MRI n = 6), 'Cannot be assessed' (11 occurrences; CT n = 8 and MRI n = 3) and 'no abnormal' (7 occurrences; CT n = 4 and MRI n = 3).

## Discussion

We found that lumbar spine X-Ray, CT and MRI reports describe a large number of findings overall. While most were judged to be of unlikely clinical relevance, less than a fifth were

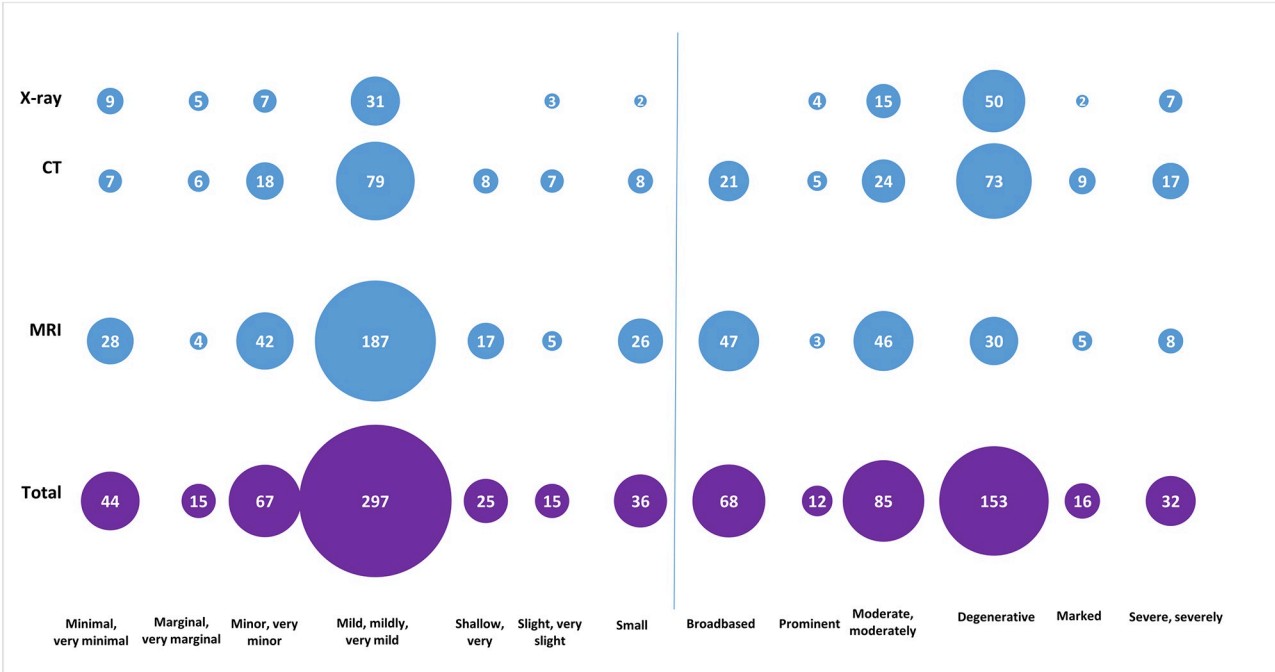

**Fig 2. Most frequent descriptors of likely clinically unimportant findings overall (used 10 or more times), and by imaging modality\*.** \*Descriptors used less than ten times overall included chronic (n = 9), focal (n = 9), subtle (n = 8), advanced (n = 7), diffuse (n = 7), significant (n = 7), generalised (n = 6), limited (n = 6), large (n = 4), considerable (n = 3), decreased (n = 3), eccentric (n = 3), fatty (n = 3), gentle (n = 3), asymmetric (n = 2), florid (n = 2), partly (n = 2), stable (n = 2), crescentic (n = 1), further (n = 1), irregular (n = 1), low level (n = 1), more substantial (n = 1), non-compressive (n = 1), occasional (n = 1), prolific (n = 1), pseudo (n = 1).

explicitly reported to be benign. Descriptors such as minimal, minor, small, mild and degenerative were common and likely intended to convey a lack of clinical relevanceVague modifiers and hedging vocabulary were also common across reports.

There was variation in the proportion of each finding reported as benign, for example, 'lumbar spine alignment' was commonly reported as normal while some terms such as spondylolisthesis were not reported as benign in any report where it was present. Terms such as spondylolisthesis are often considered 'pathological' [2] which could explain why there were less likely to be explicity identified as benign. Our findings are in keeping with the single previous study that has investigated the content of lumbar spine X-ray reports [2]. We found that these issues also apply to complex imaging (CT and MRI)reports.

Previous studies have indicated that many commonly reported, likely clinically unimportant findings, such as disc bulge and disc or facet joint degeneration, are likely to be misconstrued by patients [9, 18, 23]. An online survey designed to elicit consumer understanding of terms commonly used in lumbar spine imaging reports found these terms were deemed to be serious and likely to make the majority of respondents concerned about persistence of pain [9]. Participants in another online survey also reported lower expectations of recovery and higher perceived seriousness and need for surgery if they were deemed to have a'disc bulge' or 'degeneration' as the cause of their pain versus other diagnostic labels such as an 'episode of back pain' or 'lumbar sprain' [23].

General practitioners also want clear explanations for terms found in lumbar spine imaging reports, including their clinical relevance [24]. In another study that replaced terms considered to increase patient concern with alternative less concerning terms were interpreted as showing

**Table 3. Terms of uncertainty and their frequency overall and by imaging modality\*.**

| Terms of uncertainty | XR | CT | MRI | TOTAL |
|---|---|---|---|---|
| Not shown; Seen; identified; noted; detected; demonstrated; evident; shown; show; shows; of note | 35 | 104 | 82 | **221** |
| No significant; no suspicious; no obvious; no undue; no gross; no specific; without significant; no appreciable; no convincing; no definable; no definite; no other significant; no active | 8 | 35 | 53 | **96** |
| No evidence of; without evidence of; no features of; no signs of; without convincing evidence of; without clear evidence of; does not show any evidence of; no further evidence of; no specific evidence; with evidence of; evidence of | 10 | 30 | 39 | **79** |
| Appear; appears; apparent; appearing; in appearance; has the appearance of | 12 | 24 | 17 | **53** |
| Very mild; very slight; very minimal; very marginal; slightly; perhaps slightly; a degree of; only mildly; only very; only limited; only very minimal; looks to be slightly; there may be slight; slightly; relatively; slight; a little; limited; lower limits of | 0 | 9 | 31 | **40** |
| Significant; substantial; significantly; significance; no substantial | 2 | 8 | 20 | **30** |
| Possible; possibly; probably; potentially; potential; possibility of; perhaps | 2 | 10 | 16 | **28** |
| Suggests; suggestion of; appearances suggest; suggested; suggestion; suggestive of; suggesting; would suggest; no suggestion of | 3 | 6 | 12 | **21** |
| Some; some impression of; sometimes; somewhat | 3 | 6 | 8 | **17** |
| May; may be; may represent; may be a source; may be contributing to; may benefit from; may contribute to; may have; may just; may well respond; might be; can be | 2 | 7 | 6 | **15** |
| Likely; most likely; almost certainly; thought to reflect; this may be due to; | 3 | 3 | 7 | **13** |
| Essentially; reasonably; generally; usually; equivocal | 0 | 4 | 8 | **12** |
| Cannot be assessed; cannot be further assessed; challenging to evaluate; it is difficult to see; clinical significance unknown; as far as can be ascertained; not well demonstrated; within the limitations; I suspect; is thought to reflect; of concern | 0 | 8 | 3 | **11** |
| If the patient has clinical features of; if the patient is not benefitting; is there?; if there is; recommend clinical correlation; should that fit; favoured to related to; does this correlate clinically; if this is; in an attempt to; | 1 | 2 | 7 | **10** |
| Could be considered; could be entertained; could be performed; could pertain to; could represent; consideration should be given | 0 | 2 | 5 | 7 |
| No abnormal; no abnormality | 0 | 4 | 3 | 7 |
| Not convincing of; disproportionate to; do not necessarily; not thought to be; in the absence of; no specific cause; do not show any | 1 | 3 | 3 | 7 |
| Suspicious; appearances suspicious | 1 | 1 | 4 | **6** |
| At least; allowing for | 0 | 3 | 2 | **5** |
| Contribute; contributes; consistent with; most in keeping with; identifiable | 0 | 2 | 3 | **5** |
| Quite; quite substantial; particularly | 0 | 1 | 3 | **4** |
| Would likely account for; which would be; which would account for; would account for | 0 | 2 | 2 | **4** |

\*Terms were grouped based on similar meanings

less severe disease by general orthopaedic surgeons, orthopaedic residents and physiotherapists but not spinal surgeons [6]. Similarly, for all groups except spinal surgeons there was also a trend away from recommendations for invasive treatments such as injections and surgery, and a lower perceived likelihood of the patient requiring surgical intervention.

## Strengths and limitations

We performed a content analysis of a random sample of lumbar spine plain X-ray, CT and MRI imaging reports obtained from a large radiology service provider in Victoria, Australia, and extracted all findings until data saturation was reached for each imaging modality. It is therefore likely that our results are generalisable to other providers and settings in Australia.

Decisions regarding whether findings were likely clinically important or unimportant were made independently by two authors, based *a priori* on published evidence and report context. Equivocal findings and differences of opinion between the author team were resolved by discussion. These data are so that readers can make their own judgments about the validity of our decisions. Similarly two authors also independently determined whether or not each finding was reported to be benign and disagreements were resolved by discussion with all authors.

We were intentionally conservative in how we categorised the likely clinical importance of imaging findings. For example, we categorised moderate to severe canal stenosis as likely clinically important as this is widely accepted in clinical practice [25]. However a recent study found that, even in combination with other degenerative findings, canal stenosis may not have a clinically important association with low back pain [4]. It is also possible that some findings we coded as likely clinically unimportant were also miscoded.

The reports were extracted in full from an electronic database but we did not have access to the imaging requests other than what was included in the reports. While the clinical notes from requests are usually included in the reports in our setting, we cannot exclude the possibility that the referring clinician provided further information by other means. We were also unable to verify the report findings, or their adherence to standard lumbar spine reporting nomenclature [26], as we did not have access to the images themselves.

## Implications for practice

Although imaging is only one part of a comprehensive clinical assessment, primary care clinicians can find it difficult to understand the terminology in reports and assess the clinical relevance of findings [24]. This may lead to misinterpretation of the findings by both referring clinicians and patients and result in unwarranted anxiety, more complex imaging, overdiagnosis and unnecessary treatment. As patients are increasingly able to directly access their imaging findings, it becomes even more imperative for radiologists to consider how to report imaging findings in a way that minimises misinterpretation and uncertainty about the relevance of the reported findings.

## Implications for research

Further research is needed to determine the most effective and comprehensible methods for reporting lumbar spine imaging findings in people with low back pain. Co-design with relevant stakeholders including radiologists, clinicians and patients of a standard reporting template, that appropriately considers the importance of findings, how they are described and how certainty is qualified, is one possible approach. The template could also provide guidance about where it might be appropriate to use vague modifiers or hedging terms [16]. Evaluation of such a template could consider whether the findings are understood by clinicians and patients as intended and whether it improves the quality of care compared with usual reporting. Similar to pathology reports that provide normal ranges that can vary by age, imaging reports could also include a 'reference range' of findings that are normal for age.

However, studies that have included explicit information about the age-related prevalence of common findings in asymptomatic populations have reported conflicting results. While early studies identified promising reductions in referrals and repeat imaging [7] and reduced opioid prescription [8] a large multi-centre randomised trial involving 250,401 participants found that a small shift in prescribing only [27–29]. Changing the language to be less threatening also shows promise [6, 17] as does inclusion of explicit evidence-based management advice [30].

## Conclusion

Lumbar spine imaging reports frequently include findings that are unlikely to be clinically important without explicitly indicating they are benign. A wide variety of descriptors and uncertainty terms are used to put the findings in context that may be indirectly intended to convey the lack of clinical relevance of the findings. Clearer, more explicit language may reduce misconceptions about the relevance of lumbar spine imaging findings in people with low back pain and improve quality of care and health-related outcomes.

## Supporting information

**S1 Table. Likely clinically unimportant findings based upon the published evidence for the relevance of imaging findings\*.** \*Bracketed numbers indicate the studies that determined the findings are likely clinically unimportant.
(DOCX)

**S2 Table. Clinically unimportant findings that were reported to exclude the presence of the pathology by number of reports and overall frequency of the finding.**
(DOCX)

**S3 Table. Frequency and type of likely clinically important findings by modality.**
$^{\wedge}$N = number of reports (% of all reports for same modality). $^{\#}$N = number of times finding present (% of all findings for same modality). $^{\alpha}$N = number of times finding reported as present (% of same finding).
(DOCX)

## Acknowledgments

Thank you to Marco Polidori, our research assistant for the report extraction and to iMed radiology network, who kindly provided a copy of the anonymised reports.

## Author Contributions

**Conceptualization:** Caitlin Farmer, Romi Haas, Jason Wallis, Denise O'Connor, Rachelle Buchbinder.

**Data curation:** Caitlin Farmer, Romi Haas, Jason Wallis, Rachelle Buchbinder.

**Formal analysis:** Caitlin Farmer, Romi Haas, Jason Wallis, Denise O'Connor, Rachelle Buchbinder.

**Methodology:** Caitlin Farmer, Romi Haas, Jason Wallis, Denise O'Connor, Rachelle Buchbinder.

**Project administration:** Caitlin Farmer.

**Supervision:** Denise O'Connor, Rachelle Buchbinder.

**Visualization:** Caitlin Farmer, Denise O'Connor, Rachelle Buchbinder.

**Writing – original draft:** Caitlin Farmer, Denise O'Connor, Rachelle Buchbinder.

**Writing – review & editing:** Caitlin Farmer, Romi Haas, Jason Wallis, Denise O'Connor, Rachelle Buchbinder.

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
