## [Decision Letter · Decision Letter 0]

4 Sep 2023

PONE-D-23-14641Are clinically unimportant findings qualified as benign in lumbar spine imaging reports? A content analysis of plain X-ray, CT and MRI reportsPLOS ONE

Dear Dr. Farmer,

Thank you for submitting your manuscript to PLOS ONE. After careful consideration, we feel that it has merit but does not fully meet PLOS ONE’s publication criteria as it currently stands. Therefore, we invite you to submit a revised version of the manuscript that addresses the points raised during the review process.

We look forward to receiving your revised manuscript.

Kind regards,

Aloysius Gonzaga Mubuuke

Academic Editor

PLOS ONE

Journal Requirements:

**Additional Editor Comments:**

The paper addresses an important area trying to show how radiologists make their decisions using different imaging modalities in lumbar images. In addition to the comments from reviewers, the authors should strengthen the discussion by clearly explaining the implication of the findings to clinical practice in patients presenting with lumbar issues for imaging. In addition, proof-read the entire paper to refine the language.

Reviewers' comments:

Reviewer's Responses to Questions

**Comments to the Author**

1. Is the manuscript technically sound, and do the data support the conclusions?

Reviewer #1: Yes

Reviewer #2: Yes

2. Has the statistical analysis been performed appropriately and rigorously? 

Reviewer #1: Yes

Reviewer #2: Yes

3. Have the authors made all data underlying the findings in their manuscript fully available?

Reviewer #1: Yes

Reviewer #2: Yes

4. Is the manuscript presented in an intelligible fashion and written in standard English?

Reviewer #1: Yes

Reviewer #2: Yes

5. Review Comments to the Author

Reviewer #1: Thank you for the opportunity to review this article. The research question is if radiologists label normal findings in imaging (X-Ray, CT, and MRI) as such in their report. For this, 200 of each modality from patients in which the imaging was indicated due to lower back pain, were acquired. The reports analyzed with the question, if the normality of likely clinically unimportant findings was reported. The main finding is that only a minority of “most likely clinically unimportant” findings were not labeled as such, which, as they argue, could lead clinicians and patients to overestimate the severity of findings and lead to potentially unwarranted escalation of therapies.

The introduction clearly introduces the topic and explains the clinical relevancy in an encompassing manner. The methods are clearly described. The reporting of the is generally very detailed and transparent of the findings, although some aspects require additional explanation.

In table 1 an extra row should be added, regarding how many percent of the clinically unimportant findings were on average reported as benign. Although this is similar to “Number of reports that qualified all reported clinically unimportant findings as benign, N(%)”, it does not allow conclusions about average percent of findings that were reported as benign.

In table 2 a brief explanation should be added, that cases of potentially clinically significant findings were excluded. Although it is described in the methods, the table adds ambiguity which gradings of the reported features were included. For example, just mentioning Foraminal stenosis, might lead readers to the conclusion that all grades of foraminal stenosis were evaluated as likely clinically unimportant. The mentioning of “Lumbar spine alignment” and “Lordosis” or” Scoliosis” leads to confusion, what exactly is meant with the term “Lumbar spine alignment”.

The discussion could be improved by discussing potential reasons for the percentage differences in labeling of features as benign. For example, normal lumbar spinal alignment and Pedicles/pars defect were report as benign 100% of the time, while no occurrence “Spondylolisthesis” was reported as benign.

From an orthopedics surgeons’ point of view, the assertion that changes like Modic Typ3, Nerve root irritation, Facet joint arthropathie, or Spondylolisthesis are benign or clinically unimportant is difficult to grasp. These might be not significantly associated with low back pain on a population’s levels, as shown in some studies, but for individual patients, they can be indicative of underlying pathology that may require intervention (conservative or otherwise).

In clinical practice, a patient’s history, physical examination findings, and imaging studies must be considered together to make informed decisions on management. Furthermore, some of these changes, might be associated with mechanical instability or degenerative processes that could progress over time. As such, dismissing them as benign or clinically unimportant without a thorough evaluation and considering the patient's unique circumstances could potentially lead to missed opportunities for early intervention and management, which in turn could impact the quality of life and functional outcomes for the patient. Given that radiologist often have limited clinical data about the patient, it seems reasonable, that radiologist would not reported these findings explicitly as benign (or normal, etc.). The discussion should aim to address these concerns or better specify the population of patients this study is aimed to cover.

An additional limitation of this study that should be reported is the missing of clinical data.

Overall the manuscript is well written and answers that set out research question. After the comments are adressed, I can recommend this manuscript for publication.

Reviewer #2: Comments

Abstract line 35; qualifying that they are benign

103-104 and this study was conducted over the 104 following year

114 To ensure anonymity of the reporting radiologist, - add coma

125; serious finding is vague; quantify or substantiate on the meaning of vague.

143-146; why did physiotherapists and rheumatologists independently determined whether finding was likely clinically unimportant or important? Why not the physicians such as orthopedic surgeons or referring clinicians? Was this appropriate?

199: The 95 were excluded; Reasons for exclusion?

212; use age range instead of median age

217; use mean/average instead of median.

264; The most common finding across imaging modalities was fracture, while for CT and MRI it was foraminal and canal stenosis and nerve root impingement; So was it fracture for MRI and CT too?

297; This study found lumbar spine X-Ray, CT and MRI reports include a large number of findings.; write; This study found that; also large number of findings is vague; do you mean “irrelevant” findings or?

383; Avoid the word- serious

408: Provided; not provide.

Overall: Try to summarise This study found lumbar spine X-Ray, CT and MRI reports include a large number of findings. the discussion; it is too long.

Check all grammatical errors and correct them before submitting.

6. PLOS authors have the option to publish the peer review history of their article (what does this mean?). If published, this will include your full peer review and any attached files.

Reviewer #1: No

Reviewer #2: **Yes: **Rita Nassanga

---

## [Author Response · Author response to Decision Letter 0]

12 Dec 2023

Thank you for your constructive and considered feedback, which we have incorporated into our manuscript. We have reviewed grammar and edited for clarity, and significantly reduced the discussion length. We have also refined formatting and labelling in accordance with PlosOne policies. Please find our full response below and in the attached 'Response to reviewers' document. 

We thank the reviewers for their insightful comments on our manuscript, and answer their queries below. 

Comments to the Author

1. Is the manuscript technically sound, and do the data support the conclusions?

Reviewer #1: Yes

Reviewer #2: Yes

2. Has the statistical analysis been performed appropriately and rigorously?

Reviewer #1: Yes

Reviewer #2: Yes

3. Have the authors made all data underlying the findings in their manuscript fully available?

Reviewer #1: Yes

Reviewer #2: Yes

4. Is the manuscript presented in an intelligible fashion and written in standard English?

Reviewer #1: Yes

Reviewer #2: Yes

5. Review Comments to the Author

Reviewer #1: Thank you for the opportunity to review this article. The research question is if radiologists label normal findings in imaging (X-Ray, CT, and MRI) as such in their report. For this, 200 of each modality from patients in which the imaging was indicated due to lower back pain, were acquired. The reports analyzed with the question, if the normality of likely clinically unimportant findings was reported. The main finding is that only a minority of “most likely clinically unimportant” findings were not labeled as such, which, as they argue, could lead clinicians and patients to overestimate the severity of findings and lead to potentially unwarranted escalation of therapies.

The introduction clearly introduces the topic and explains the clinical relevancy in an encompassing manner. The methods are clearly described. The reporting of the is generally very detailed and transparent of the findings, although some aspects require additional explanation.

In table 1 an extra row should be added, regarding how many percent of the clinically unimportant findings were on average reported as benign. Although this is similar to “Number of reports that qualified all reported clinically unimportant findings as benign, N(%)”, it does not allow conclusions about average percent of findings that were reported as benign.

We have added a row in Table 1, “Mean percent of clinically unimportant findings described as benign per report” as requested. We also reordered one row on the table. 

In table 2 a brief explanation should be added, that cases of potentially clinically significant findings were excluded. Although it is described in the methods, the table adds ambiguity which gradings of the reported features were included. For example, just mentioning Foraminal stenosis, might lead readers to the conclusion that all grades of foraminal stenosis were evaluated as likely clinically unimportant. The mentioning of “Lumbar spine alignment” and “Lordosis” or” Scoliosis” leads to confusion, what exactly is meant with the term “Lumbar spine alignment”.

Thank you for this feedback. We have added descriptors to the relevant terms (for example, ‘Scoliosis (mild)’ to make this clearer to readers, and added a note regarding alignment terms, which were reproduced as they were written in the report. 

The discussion could be improved by discussing potential reasons for the percentage differences in labeling of features as benign. For example, normal lumbar spinal alignment and Pedicles/pars defect were report as benign 100% of the time, while no occurrence “Spondylolisthesis” was reported as benign.

We have revised the discussion and added a paragraph regarding the potential reasons for the range in labelling of findings as benign as suggested (lines 346-350 in track changes manuscript). 

From an orthopedics surgeons’ point of view, the assertion that changes like Modic Typ3, Nerve root irritation, Facet joint arthropathie, or Spondylolisthesis are benign or clinically unimportant is difficult to grasp. These might be not significantly associated with low back pain on a population’s levels, as shown in some studies, but for individual patients, they can be indicative of underlying pathology that may require intervention (conservative or otherwise).

In clinical practice, a patient’s history, physical examination findings, and imaging studies must be considered together to make informed decisions on management. Furthermore, some of these changes, might be associated with mechanical instability or degenerative processes that could progress over time. As such, dismissing them as benign or clinically unimportant without a thorough evaluation and considering the patient's unique circumstances could potentially lead to missed opportunities for early intervention and management, which in turn could impact the quality of life and functional outcomes for the patient. Given that radiologist often have limited clinical data about the patient, it seems reasonable, that radiologist would not reported these findings explicitly as benign (or normal, etc.). The discussion should aim to address these concerns or better specify the population of patients this study is aimed to cover.

We agree that it is important to comprehensively assess an individual patient and we would expect that the GP/primary care practitioner would do so. We have made it clearer that in primary care most of these findings would be considered benign. Our judgements about relevance of individual findings were based upon the available published evidence that these findings are usually benign – as we indicated, both cross-sectional and longitudinal population-based studies have found that the changes we included as benign do not appear to be associated with either current or future back pain. 

This is outlined in the methods (lines 143-158) and discussion (lines 375-381). In the following paragraph in the discussion (lines 384-389) we also indicated that we were intentionally conservative in our judgements. As well we provided these data in full (Supplementary Table 1) so that the reader can also make their own judgments about our assessments. We agree that our judgements were based upon population-based data, which is the highest form of evidence for looking at these types of associations and together with clinical judgement can be therefore extrapolated to individual patients in most instances. We agree that in a specialist setting, the specialist can review the films and make their own interpretation for individual patients. 

An additional limitation of this study that should be reported is the missing of clinical data.

We have added a sentence clarifying this (lines 392-394). 

Overall the manuscript is well written and answers that set out research question. After the comments are adressed, I can recommend this manuscript for publication.

Thank you for these helpful comments.

Reviewer #2: Comments

Abstract line 35; qualifying that they are benign 

Corrected, thank you.

103-104 and this study was conducted over the following year 

Added as requested.

114 To ensure anonymity of the reporting radiologist, - add coma 

Added as requested.

125; serious finding is vague; quantify or substantiate on the meaning of vague. 

We have expanded the examples of serious findings.

143-146; why did physiotherapists and rheumatologists independently determined whether finding was likely clinically unimportant or important? Why not the physicians such as orthopedic surgeons or referring clinicians? Was this appropriate? 

Findings were determined as likely clinically unimportant or important based on the available evidence (Supplementary table 1). The research authors were physiotherapists and a rheumatologist. Their background was included as context about who was performing the study and the judgements. All are expert clinicians with expertise in back pain so we consider this appropriate. 

199: The 95 were excluded; Reasons for exclusion? 

We have clarified in the text that the reasons for exclusion are provided in Figure 1 (top box, right side of figure). 

212; use age range instead of median age 

We prefer to report both median and range for age. 

217; use mean/average instead of median. 

We prefer to report median rather than mean as the data were not normally distributed. 

264; The most common finding across imaging modalities was fracture, while for CT and MRI it was foraminal and canal stenosis and nerve root impingement; So was it fracture for MRI and CT too? 

We have clarified that the most common finding for x-ray was fracture, for CT it was canal stenosis, and for MRI it was foraminal stenosis.

297; This study found lumbar spine X-Ray, CT and MRI reports include a large number of findings.; write; This study found that; 

We have amended this as suggested.

also large number of findings is vague; do you mean “irrelevant” findings or? 

We have clarified this sentence to “This study found that lumbar spine X-Ray, CT and MRI reports describe a large number of imaging findings overall.” 

383; Avoid the word- serious 

We prefer to retain this word as it is based upon its use in clinical guidelines and our prior cited papers. 

408: Provided; not provide. 

Amended as suggested.

Overall: Try to summarise This study found lumbar spine X-Ray, CT and MRI reports include a large number of findings. the discussion; it is too long.

Check all grammatical errors and correct them before submitting.

We have shortened the discussion and checked and corrected grammatical errors.

---

## [Editor Report · Decision Letter 1]

16 Jan 2024

Are clinically unimportant findings qualified as benign in lumbar spine imaging reports? A content analysis of plain X-ray, CT and MRI reports

PONE-D-23-14641R1

Dear Dr. Farmer,

We’re pleased to inform you that your manuscript has been judged scientifically suitable for publication and will be formally accepted for publication once it meets all outstanding technical requirements.

Kind regards,

Aloysius Gonzaga Mubuuke

Academic Editor

PLOS ONE
---

## [Editor Report · Acceptance letter]

4 Mar 2024

PONE-D-23-14641R1 

PLOS ONE

Dear Dr. Farmer, 

I'm pleased to inform you that your manuscript has been deemed suitable for publication in PLOS ONE. Congratulations! Your manuscript is now being handed over to our production team.

Kind regards, 

on behalf of

Dr. Aloysius Gonzaga Mubuuke 

Academic Editor

PLOS ONE